# A data-driven wind-to-current response function and application to Ocean surface current estimates

Clément Ubelmann[1], J. Thomas Farrar[2], Bertrand Chapron[3], Lucile Gaultier[4], Laura Gomez-Navarro[5,6], Marie-Hélène Rio[7], and Gérald Dibarboure[8]

[1]Datlas, 70 Rue de la Physique, 38400 Saint-Martin-d'Hères, France
[2]Woods Hole Oceanographic Institution, 266 Woods Hole Road, Woods Hole, MA 02543-1050 U.S.A
[3]Ifremer, LOPS, UMR CNRS 6523, 29280 Plouzané, France
[4]Ocean Data Lab, 29280 Plouzané, France
[5]Utrecht University, Heidelberglaan 8, 3584 CS Utrecht, Netherlands
[6]Institut de Ciencies del Mar (CSIC), Pg. Marítim de la Barceloneta, 37, Ciutat Vella, 08003 Barcelona, Spain
[7]European Space Agency, ESA-ESRIN, Via Galileo Galilei, 1, 00044 Frascati RM, Italy
[8]Centre National d'Etudes Spatiales, 18 Av. Edouard Belin, 31400 Toulouse, France

**Correspondence:** Clément Ubelmann (clement.ubelmann@datlas.fr)

**Abstract.** This study investigates the reconstruction of wind-driven currents based on an empirical impulse response function. Surface current observations derived from drifting buoy data and wind-stress from the ERA5 reanalyses are used to derive the response function. The function is expected to be sensitive to the ocean mixed-layer depth and more generally the turbulent viscosity profile which can display strong spatio-temporal variability. In this work, however, only seasonal and meridional variations are considered. Despite this crude approximation, the simplified response function can explain a significant portion of the current variability in independent observations.

A practical application is the release of a new total surface current product (denoted WOC). Compared to existing products based on the same input datasets, such as the CMEMS MOB-TAC (Guinehut, 2021) surface current product, the WOC estimates are designed to include higher frequency content, in particular in the inertial band. Beyond successful validation, the characteristics of the response function (amplitudes and phases) reveal interesting properties of the upper-ocean variability. The function shows some similarities to one derived theoretically from a simple 1-layer (slab) model, but also differences that highlight the value of fitting the function to the data without the use of an explicit dynamical model. These results open perspectives for studying some dependencies between subsurface variables and the response function, particularly interesting in the context of future spaceborne Doppler scatterometers such as ODYSEA (Rodríguez et al., 2019), expected to provide simultaneous wind and current observations. This instrument could indirectly probe subsurface properties through the synoptically-observed response function.

## 1  introduction

The transfer of momentum and energy across the air-sea interface provides sources of oceanic motion. The resulting upper ocean surface currents can then cover a wide range of temporal and spatial scales. A major component, called the geostrophic

current, equilibrates the pressure gradient force and the Coriolis force. Pressure gradients are currently well observed by satellite altimetry at spatial and temporal scales down to about 150km wavelength and 20-day periods (Ballarotta et al., 2019). Another important component, called wind-driven current, is more directly related to atmospheric wind stress forcing. This includes both Ekman currents, which result from a balance of the "frictional" force (the wind stress at the surface and subsurface turbulent momentum flux) and the Coriolis force, and inertial currents, which result from the resonant response of the upper ocean to changing winds. These currents are considered as ageostrophic as they are a departure from the geostrophic equilibrium. Wind-driven currents can reach large amplitudes, often exceeding the geostrophic current. They play an important role in the energy budget of the ocean (Flexas et al., 2019) and are of great interest for practical and societal applications. One example is surface drift and accumulation of marine litter (Higgins et al., 2020; Cunningham et al., 2022). Besides seldom satellite synthetic aperture radar estimates (Chapron et al., 2005), total surface currents are not directly captured at synoptic scale by satellite observations. However, estimates can be obtained from knowledge of the surface wind stress. In this study, we investigate the use of a data-driven response function relating the wind-stress and the ageostrophic surface current to empirically capture some part of the wind-driven currents.

Some recent studies have been dedicated to the theoretical aspects of the response of upper-ocean currents to wind forcing. In particular, Elipot and Gille (2009) and Lilly and Elipot (2021) focused on the spectral transfer function between wind-stress and current, with extensive analyses of its dependencies on viscosity profiles as a function of depth. We focus here on the closely related impulse response function, or just response function, that relates the ocean response to the wind forcing in the physical space. The impulse response function is the Fourier transform of the spectral transfer function (Bendat and Piersol, 2010, p. 29, 26-27). The construction of the response function from real data and its applications to estimate the surface current at synoptic scales have not been fully explored yet. Existing operational surface current products include an estimation of ageostrophic current related to wind forcing, such as the [OSCAR] or the [CMEMS-MOB-TAC] datasets also based on a response function as described in Rio et al. (2014). Their response function from wind-stress to surface current is a single complex-scalar function therefore responding equally to all frequencies, designed to empirically capture Ekman currents.

To generalize the approach and, in particular, to better resolve the inertial frequency band, here we examine the empirical fit of a full response function acting across a broader spectral range. As detailed in Lilly and Elipot (2021), the local response of the ocean to wind forcing at different frequencies can be described with a complex frequency response function, which is equivalent to use of a complex impulse response function in physical space. In this study, we therefore propose to explore the empirical fit of a convolution response function and show its ability to reconstruct some ageostrophic surface current directly related to wind forcing. This is made possible thanks to the growing number of accumulated drifter data at high temporal frequency (hourly outputs). One practical application is the estimation of some wind-driven surface current directly from the available wind-stress reanalysis products. Also, a more exploratory objective is to analyze whether the empirical response function constructed from the data alone can help us obtain new insights into ocean physics (like vertical mixing) and subsurface ocean properties (like mixed-layer depth). This is strongly motivated by the prospect of future spaceborne Doppler missions such as ODYSEA (Rodríguez et al., 2019) designed to observe simultaneously the surface wind and current

at synoptic scale. Indeed, if the sparse drifter database can only provide spatio-temporally averaged response functions at best,
the space-borne observation may allow a monitoring of the response function to probe subsurface characteristics.

This manuscript is organized as follows : section 2 presents all the datasets used in this study, as input or validation datasets. Then, section 3 focuses on the methodology behind the response function. Section 4 covers the application to surface current estimates, including the validation, and section 5 explores some characteristics of the response function and its characteristics with respect to subsurface dynamics. Finally, section 6 concludes and discusses some perspectives.

## 2   Datasets

The empirical fit, performed globally over 70°S to 80°N, is based on three input datasets: the surface drifter velocities (sparse total current observations), the geostrophic velocities (to estimate the ageostrophic current by difference with the total current), and the wind stress from the ERA5 reanalysis, all covering the period from years 2010 to 2020. An additional dataset of total surface current, based on similar input datasets but using a different algorithm, is considered for comparison to our total surface current estimates.

The surface drifter velocities have been extracted from the Global Drifter Program [GDP] database (Elipot et al., 2016) in its version 2.0.1. Only the 'drogued' drifters are considered in the main experiment, representative of the current at 15m depth which is the focus of this study. The velocities at hourly frequency are used (estimated jointly from the unevenly distributed observed positions). Both ARGOS and GPS data are considered to allow the 10-year extension of the study with a maximum number of data, although the GPS data, collected with a different technology, are more accurate (Yu et al., 2019) and fairly dominant after 2015.

The geostrophic velocities used in this study were derived from muti-satellite altimetry maps (Taburet et al., 2019). The data, already processed in velocity units (m/s), were extracted from the [CMEMS-MOB-TAC] dataset. We co-located the data at all drifters hourly positions. A linear interpolation scheme was used between the daily 1/4° spatial grid and the drifter positions. By difference with the total current oberved from the drifters, we have an estimation of the ageostrophic component representative of the $u_e$ variable in the equations presented next section, at all drifter positions.

The surface wind-stress data were extracted from the [ERA5] product provided by the Copernicus Marine service. The time resolution is hourly and the spatial resolution is 0.25° in longitude and latitude (Hersbach et al. (2020)). We also co-located these hourly data at all drifters position, including the 8-day history in order to integrate the $\tau_0$ variable in the equations presented next section.

For validation purposes, the total surface current from the CMEMS-MOB-TAC dataset have also been used and co-located at the drifter positions.

Finally, our WOC output dataset (the acronym stands for the ESA "World Ocean Circulation" project) presented in this study, arising from the first three datasets, can be accessed here: [WOC]. The data have been written on the same grid as the total surface current from CMEMS-MOB-TAC to facilitate comparisons. Note that both the total surface current from CMEMS MOB-TAC and WOC have the same geostrophic component.

Figure 1 illustrates the input and comparison datasets, during an event where a strong atmospheric front resolved by ERA5 seems to trigger inertial currents captured by a drifter. On the upper-right panel, the drifter features clear oscillations after crossing the atmospheric front. The oscillations are very clear both on the drifter trajectory and on the derived zonal current shown on the bottom panel. Although the oscillations may combine several effects possibly including tidal signals, they are mostly inertial signal (matching well the inertial frequency at 45°N) that could be reconstructed from the wind forcing. This gives some confidence on the reliability of the datasets to explore the wind-driven current response, as well as all the previous studies on wind driven currents based on drifters. The geostrophic current shown in green explains a large part of the low-frequency motion not directly related to local wind-forcing. The CMEMS MOB-TAC total surface current that will be our baseline for comparison, shown in blue, seems to capture some accurate ageostrophic current (beyond the geostrophic one) but not the oscillatory part.

## 3 The data-driven response function

### 3.1 The rationale for a response function

The equations governing the horizontal currents in the upper ocean can be written (neglecting horizontal advection) as (e.g., Gill, 1982, p. 320):

$$\frac{\partial u_x}{\partial t} - f u_y = \frac{1}{\rho} \left( -\frac{\partial p}{\partial x} + \frac{\partial \tau_x}{\partial z} \right) \tag{1}$$

$$\frac{\partial u_y}{\partial t} + f u_x = \frac{1}{\rho} \left( -\frac{\partial p}{\partial y} + \frac{\partial \tau_y}{\partial z} \right) \tag{2}$$

where $(u_x, u_y)$ is the horizontal current vector, $f$ is the local signed Coriolis parameter, $\rho$ is the density, $p$ is the pressure and $(\tau_x, \tau_y)$ is the horizontal stress vector. All variables except $f$ are depth dependent. Assuming there is no nonlinear dependence of $(\tau_x, \tau_y)$ or $p$ on $(u_x, u_y)$ these equations are linear. Some simple parameterizations of the momentum fluxes $(\tau_x, \tau_y)$ in terms of the velocity are linear (e.g., constant eddy viscosity, linear drag), but more complicated ones are not (e.g., mixing schemes that involve a critical Richardson number). In reality, we expect a nonlinear relationship between $(\tau_x, \tau_y)$ and $(u_x, u_y)$, as well influence of other factors (e.g., surface heat fluxes), but we will assume a linear relationship as a starting point here.

We are interested in how the upper ocean responds to wind forcing. We can conceptually separate the velocity vector $(u_x, u_y)$ into a pressure-driven component and a stress-driven component $(u_{e_x}, u_{e_y})$, which is governed by:

$$\frac{\partial u_{e_x}}{\partial t} - f u_{e_y} = \frac{1}{\rho} \frac{\partial \tau_x}{\partial z} \tag{3}$$

$$\frac{\partial u_{e_y}}{\partial t} + f u_{e_x} = \frac{1}{\rho} \frac{\partial \tau_y}{\partial z} \tag{4}$$

It is convenient to to express the vectors $\mathbf{u_e} = (u_{e_x}, u_{e_y})$ and $\boldsymbol{\tau} = (\tau_x, \tau_y)$ using complex notation as $\mathbf{u_e} = u_{e_x} + i u_{e_y}$ and $\boldsymbol{\tau} = \tau_x + i\tau_y$ (where $i = \sqrt{-1}$). Then Eqns. 3-4 can be written in a single equation:

$$\frac{\partial \mathbf{u_e}}{\partial t} + i f \mathbf{u_e} = \frac{1}{\rho} \frac{\partial \boldsymbol{\tau}}{\partial z} \tag{5}$$

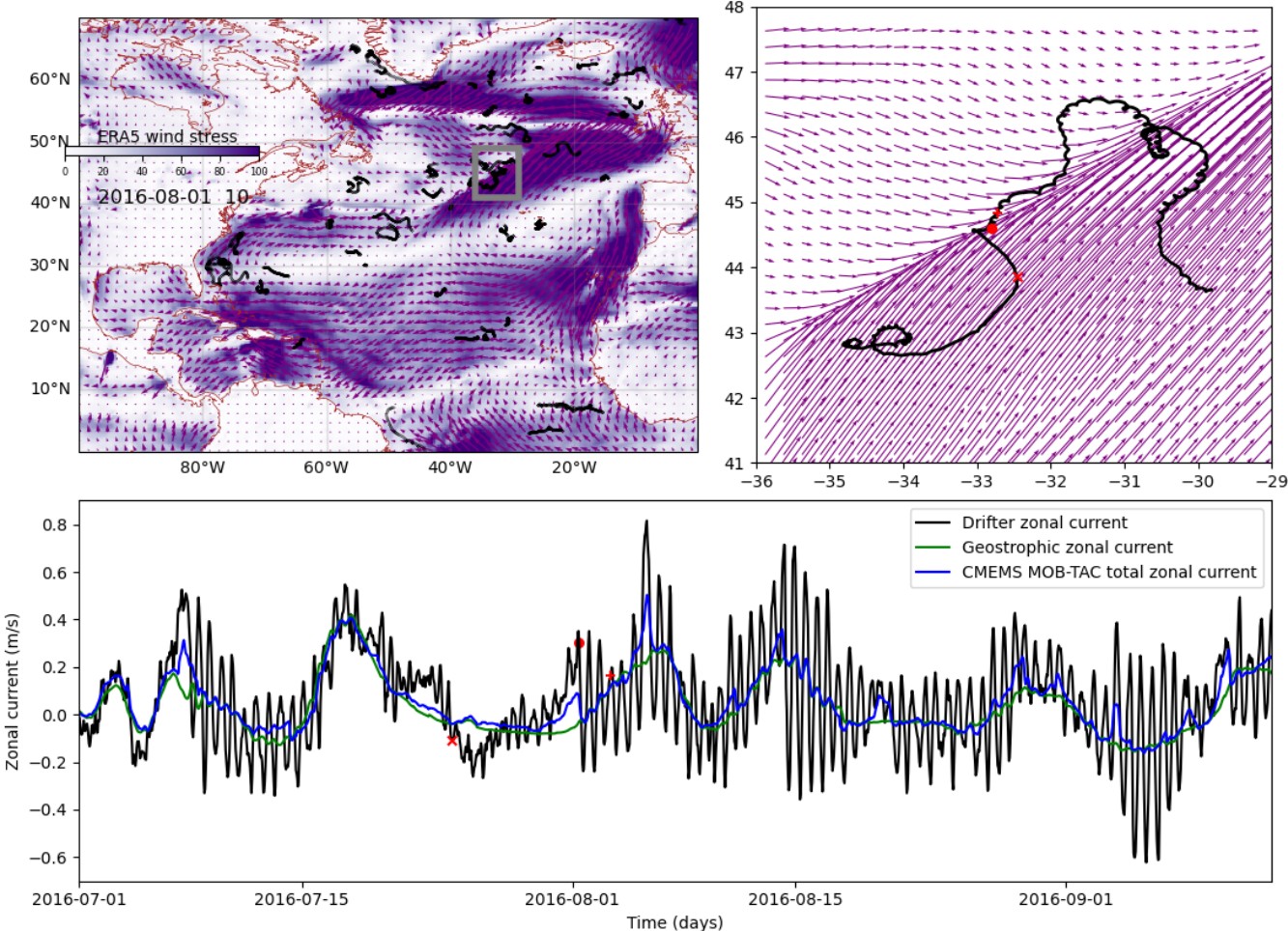

**Figure 1.** Illustration of the main datasets used in the study. Upper-left panel : a snapshot of the ERA5 wind stress superimposed with the ensemble of drifter positions over +/- 20 days. Upper right: zoom of the first panel highlighting the presence of a drifter near a strong atmospheric front (the red dot is the position at the time of the wind-stress map, the red "x" and "+" 8 days before and 2 days after respectively). Lower panel: time series of the zonal velocities derived from the drifter trajectory (black), with a colocation of the geostrophy (green) and the total surface zonal current from CMEMS-MOB-TAC (blue).

The impulse response function provides a useful way of characterizing a constant-parameter linear system and relating its inputs to its outputs. For any arbitrary input forcing at the surface, $\boldsymbol{\tau_0}(t)$, the output of the system, $\boldsymbol{u_e}(z,t)$ at depth $z$ can be written,

$$\boldsymbol{u_e}(z,t) = \int_0^T \boldsymbol{G_z}(t')\boldsymbol{\tau_0}(t-t')dt' \tag{6}$$

where $G_z$, the impulse response function of the system at depth $z$, is a complex function of time lag $t'$, and the integral from 0 to $T$ (positive time only) expresses the fact that the output $u_e$ can only depend on the past forcing $\tau$ ($t' > 0$). If we assume that the wind-driven current is only affected by the wind history over a limited time (before momentum fluxes dissipate the upper layer energy), we might choose $T$ to be on the order of a few days. As discussed in the next section, $T = 8$ days will be a reasonable value.

To get some intuition for the kinds of physics that might be captured by an empirically estimated impulse response function, it is helpful to consider a simplified model. Vertically integrating Eqn 5 from the surface to some depth $H$, the vertically averaged velocity $\bar{u}_e$ is expressed as:

$$\frac{\partial \bar{u}_e}{\partial t} + i f \bar{u}_e = \frac{\tau_0 - \tau_H}{\rho H} \tag{7}$$

where $\tau_H$ is the value of the turbulent stress vector at depth $H$. This equation is one version of the "slab model" that is commonly used to model mixed-layer inertial currents (e.g., Plueddemann and Farrar, 2006; Alford, 2020). If we parameterize the stress at depth $H$ as being linearly proportional to the layer-averaged velocity, so that $\tau_H = r\rho H \bar{u}_e$, where $r$ is a scalar damping coefficient, we obtain the well-known "damped slab" model of the mixed layer (e.g., D'Asaro, 1985):

$$\frac{\partial \bar{u}_e}{\partial t} + (r + if)\bar{u}_e = \frac{\tau_0}{\rho H} \tag{8}$$

We can derive the spectral transfer function by taking the Fourier transform of Eq. 8 (with $\bar{U}_e(\omega)$ and $T_0(\omega)$ indicating the Fourier transforms of $\bar{u}_e(t)$ and $\tau_0(t)$):

$$\bar{U}_e(\omega) = \frac{1}{\rho H(r + i(\omega + f))} T_0(\omega) \tag{9}$$

which has the impulse response function in the physical space:

$$G_0(t) = \frac{e^{-rt}}{\rho H} e^{-ift} \tag{10}$$

$G_0(t)$ is defined for $t = 0$ to $t = \infty$. For the damped slab model, the impulse response function oscillates at frequency $f$ with an amplitude that is inversely proportional to mixed-layer depth, $H$, and decays with time with an e-folding decay timescale of $1/r$. This example of $G$ function will serve as a baseline for comparison with the empirical $G$ fitted from the data in this study, and possible departures from it may reveal various kinds of additional physics that cannot be described by a single damping parameter.

## 3.2 Resolution of the inverse problem to fit G

The inversion problem consists of finding the $G_z$ function at depth $z = 15$m (noted $G$ in the following) from drifter observations $u_{obs}$, the co-located geostrophic current $u_{g_{obs}}$ and the surface stress $\tau_0$ such as:

$$u_{obs} - u_{g_{obs}} = \int_0^T G(t')\tau_0(t - t')dt' + \epsilon \tag{11}$$

$(\boldsymbol{u_{obs}} - \boldsymbol{u_{g_{obs}}})$, noted $\boldsymbol{u_{e_{obs}}}$ in the following, represents our best observed estimate of ageostrophic current, that is supposed to contain the linear response to wind forcing $\boldsymbol{\tau_0}(t)$ plus additional signal represented by $\epsilon$. $\epsilon$ may contain errors in $\boldsymbol{u_{g_{obs}}}$, errors in the drifter measurement of current, the result of error of $\boldsymbol{\tau_0}(t)$ and any ageostrophic current that would not be captured by the convolution of $\boldsymbol{G}$ with the forcing $\boldsymbol{\tau_0}(t)$. Note that $\epsilon$ is not necessarily small, but this should not prevent to find a meaningful $\boldsymbol{G}$ function if a large amount of observations are processed.

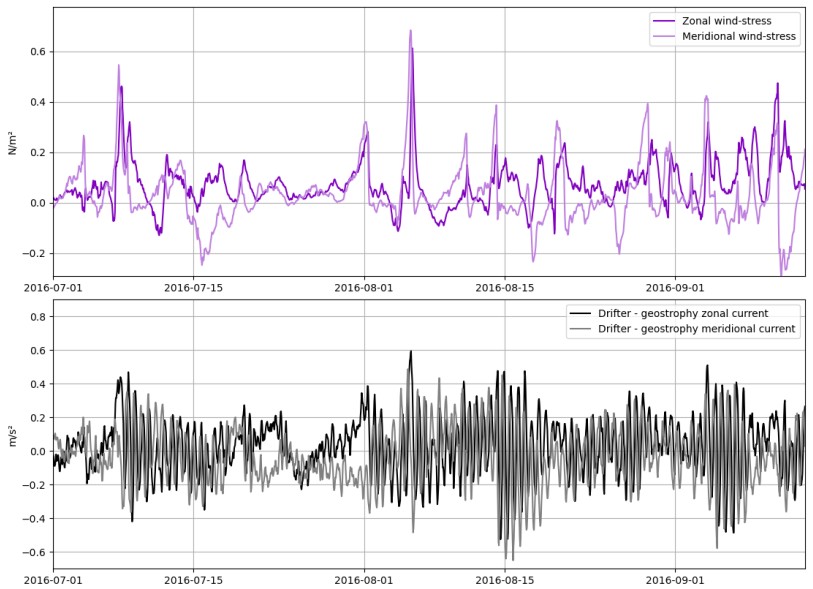

**Figure 2.** Example of Lagrangian time series of the ERA5 wind stress (zonal and meridional components) co-located at a drifter position (upper panel) and time series of the drifter ageostrophic velocity (lower panel).

As an illustration, Figure 2 shows some time series of the forcing $\boldsymbol{\tau_0}(t)$ (upper panel) and the ageostrophic observed current $\boldsymbol{u_{e_{obs}}}$ (lower panel). Solving Eq. 11 consists in finding the convolution operator transforming the upper panel series into the lower panel series, both written under the complex mathematical form. Here only 75 days of data is shown for one specific drifter, but the whole series over 2010-2020 are considered.

Finding $\boldsymbol{G}$ that minimizes $\epsilon$ in Eq. 11 is a linear inverse problem that can be solved by minimizing the following cost function :

$$J = \| \int_0^T \boldsymbol{G}(t')\boldsymbol{\tau_0}(t-t')dt' - \boldsymbol{u_{e_{obs}}} \|^2 \tag{12}$$

Over the oceans, very different conditions of stratification (and mixed layer depth in particular) can be found so we cannot expect the $\boldsymbol{G}$ response function to be uniform. Nevertheless, the amount of drifter data is limited and to avoid over-fitting issues, we cannot let $\boldsymbol{G}$ vary totally freely. In order to have a good compromise, we defined a reduced space where the $\boldsymbol{G}$

function can vary with latitude and seasons, which seemed to be the dominant variables. The impact of these assumptions on potential weaknesses of the method will be discussed in the conclusion section. In practice, we choose 1° latitudinal steps and a single harmonic (defined by 3 parameters) at 1-year period for the time variations of $G$. If $\boldsymbol{\eta}$ is a parameter vector in this reduced space, $\boldsymbol{G}$ is decomposed by a series of linear operators under the form:

$$\boldsymbol{G}(y,t,t') = \boldsymbol{\Gamma}(t')\boldsymbol{S}(t)\boldsymbol{L}(y)\boldsymbol{\eta} \tag{13}$$

where $\boldsymbol{L}$ is a bi-linear spatial interpolator transforming the ensemble of values of $\boldsymbol{\eta}$ in the parameter space into a local set of parameters at latitude $y$. Then, the operator $\boldsymbol{S}(t)$ applies the the 1-year harmonic (in practice, one constant, one sine and one cosine functions are defined at the annual-frequency). Finally, $\boldsymbol{\Gamma}$ converts the subset of parameters into the response function $\boldsymbol{G}(t')$. The number of parameters (size of $\boldsymbol{\eta}$) to fit is directly proportional to the time window over which $\boldsymbol{G}$ is defined. Some sensitivity tests have been conducted to find an optimal time extension, based on the maximum of explained variance over independent drifter data. Globally, the optimal was around 8 days, which is certainly a compromise between the theoretical extension of $\boldsymbol{G}$ (the wind-driven linear response time) and possible overfitting due to the limited amount of drifter data. Note that this optimal value may actually vary with latitudes, but we did not implement this capability.

The series of operators that transform $\boldsymbol{\eta}$ into the local (spatially and seasonally) convolution function are linear. The convolution operator is also linear. Therefore, observations at the drifter location can be written as $\boldsymbol{u}_{e_{obs}} = \boldsymbol{M}\boldsymbol{\eta} + \boldsymbol{\epsilon}$ where $\boldsymbol{M}$ is the linear operator including the successive construction of $\boldsymbol{G}$ and the integration operation with the wind stress, all linear with respect to $\boldsymbol{\eta}$ .

The cost function in Eq. 12 becomes:

$$J = \|\boldsymbol{M}\boldsymbol{\eta} - \boldsymbol{u}_{obs}\|^2 \tag{14}$$

that can be easily solved with a conjugate gradient descent involving iterative computations of the gradient of the cost function:

$$\nabla J = \frac{1}{2}\boldsymbol{M}^T(\boldsymbol{M}\boldsymbol{\eta} - \boldsymbol{u}_{obs}) \tag{15}$$

In practice, the computation of $\nabla J$ does not involve the explicit writing of the adjoint matrix $\boldsymbol{M}^T$. An operator function $\boldsymbol{M}^T$ is applied, based on the adjoint of the linear operations in Eq. 13 and the adjoint of the convolution Eq. 6. For the problem considered, the convergence was reached after about a hundred iterations with the Newton-CG scipy.optimize library in python.

## 4 Application to surface current estimates and validation

A direct application of the response function fitted from the drifters is an estimation of the linear response part of the wind-driven current (our WOC estimate). This was carried out over the 10 years of the study on the 0.25 ° resolution grid of the ERA5 input dataset. The upper panels of figure 3 show snapshots of the WOC current compared to the current from the CMEMS MOB-TAC (left). On the right, higher amplitudes are reached, with an imprint of spatial oscillatory patterns after the crossing of the atmospheric front near 45°N, 40°E. The lower panel shows these estimations as a function of time in red and blue,

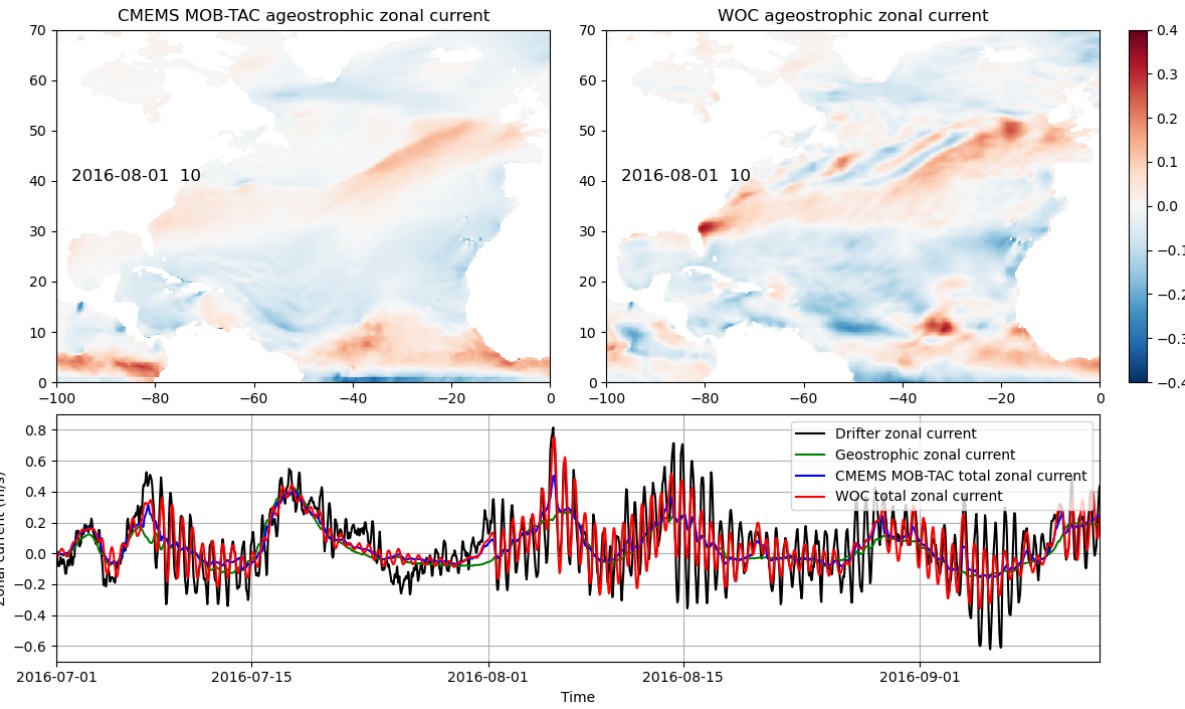

**Figure 3.** Snapshots of the ageostrophic zonal current from the CMEMS MOB-TAC (upper left) and from the WOC (upper right). Bottom : time series of the total zonal current measured by an independent drifter (black), with a co-location of the geostrophic, CMEMS MOB-TAC total and WOC total zonal current in green, blue and red respectively

respectively (with added geostrophy represented in green) co-located with a drifter in black (this drifter was excluded from the training).

A significant part of the observed ageostrophic current is captured by the WOC response function estimation (about 50% of the variance in the example shown in figure 3). The estimated near-inertial oscillations seem to be reconstructed with a phase evolution that is quite accurate in this example. (We picked a case with a particularly intense inertial signal for illustration.) The amplitude is attenuated with respect to observations, presumably because of the unresolved processes mentioned in the previous section.

Some quantitative diagnostics can be applied to the ensemble of independent drifters to assess the reconstruction skills more quantitatively and in all situations (not only during strong wind events). On the top panels of figure 4, we represent in black the power time-frequency spectrum of the observed drifter current between 1000 hours and 2 hours (in the clockwise direction on the left panel and counter-clockwise on the right panel) averaged over the oceans between 40°N and 50°N. The thick colored lines represent the resolved energy by the different estimations: geostrophic in green, total current from CMEMS MOB-TAC in

blue, and the WOC estimation in red. As anticipated by the resolved oscillations on Figure 3, the red spectrum features a clear peak at the inertial frequency (near 18 hours at these latitudes in the clockwise panel corresponding to anticyclonic motion),

of about 40% of the energy seen by the drifters (black) at the inertial frequency. We note that the sub-inertial band between 100 hours and 18 hours has also gained some energy compared to the CMEMS MOB-TAC product. However, it is interesting to note that the counter-clockwise spectrum is very similar to CMEMS MOB-TAC, and only slightly above the spectrum of geostrophic current. The second peak at 12h frequency, present in both clockwise and counter-clockwise spectra of the drifter, is not resolved by any of the estimates. It corresponds to tidal currents (barotropic near the continental shelves, and mostly baroclinic in open-ocean) not resolved by design of the different products.

The levels of energy do not tell us anything about whether the reconstructions have accurate phases. To examine the accuracy of the phase, we also computed the spectrum of the observations minus the spectrum of the difference between the estimation and the observations. This diagnostic shows how much of the observation variance is explained by the estimation (the thin colored lines). We note that overall the levels are similar to the spectra of the estimations, suggesting that the phases of the resolved signals are correct. One exception to this is the CMEMS MOB-TAC estimation in the inertial band: the energy is very low (no inertial peak), but the explained variance is significant, suggesting that the phases are correct although the energy is damped. This is consistent with what we can observe on Figure 3: the blue curve tends to follow the first oscillation of NIO events, but with a strong attenuation and only immediately after the wind impulses (by design of the non-convolutive response function).

The resolved variances are also represented in percentages on the bottom-left panel of Figure 4. Not surprisingly, the WOC with its inertial component captures more energy in the near-inertial band (30%-40% more), confirming the qualitative results from Figure 3. Also, at lower frequencies, the skill scores are similar between the CMEMS MOB-TAC and WOC (bringing a slight improvement beyond geostrophy). However, there is still 40% to 60% variance of the current missing in the sub-inertial to inertial frequency range.

Regarding the counter-clockwise scores (cyclonic), the percentages suggest that the CMEMS MOB-TAC and WOC are fairly similar, with slight improvements compared to geostrophy.

The same diagnostics have been performed in the tropical region between 5°-10°N as shown on Figure 5. In this region, the inertial frequency is low (spread between 200 and 50 hours) so different types of dynamics may coexist in the inertial band. Nevertheless, the peak of energy is clear over the inertial band and the reconstruction skills are comparable to that of the higher latitudes. We note that the WOC spectrum drops more rapidly in the super-inertial band, but where none of the products have significant scores above zero anyway (the phases are not consistent with observations in the super-inertial band). This suggests that we are not resolving surface currents at short time scales in the Tropics, and possible diurnal or semi-diurnal effects are not captured, as discussed later in the conclusion section.

Regarding the counter-clockwise scores (cyclonic), the results are also similar to that of mid-latitudes (with overall less contribution from the geostrophic estimate as expected in the tropics).

The explained variances as a function of latitude is represented in Figure 6. In this diagnostic, all frequencies are considered, but the view along the latitude dimension, separately for the zonal and meridional current, is instructive. We note the strong zonal current variability of the Equatorial currents seen by the drifters. Here, the altimetry contribution is actually the extension of geostrophy based on the Lagerloef et al. (1999) derivation implemented in the CMEMS geostrophic current product near the

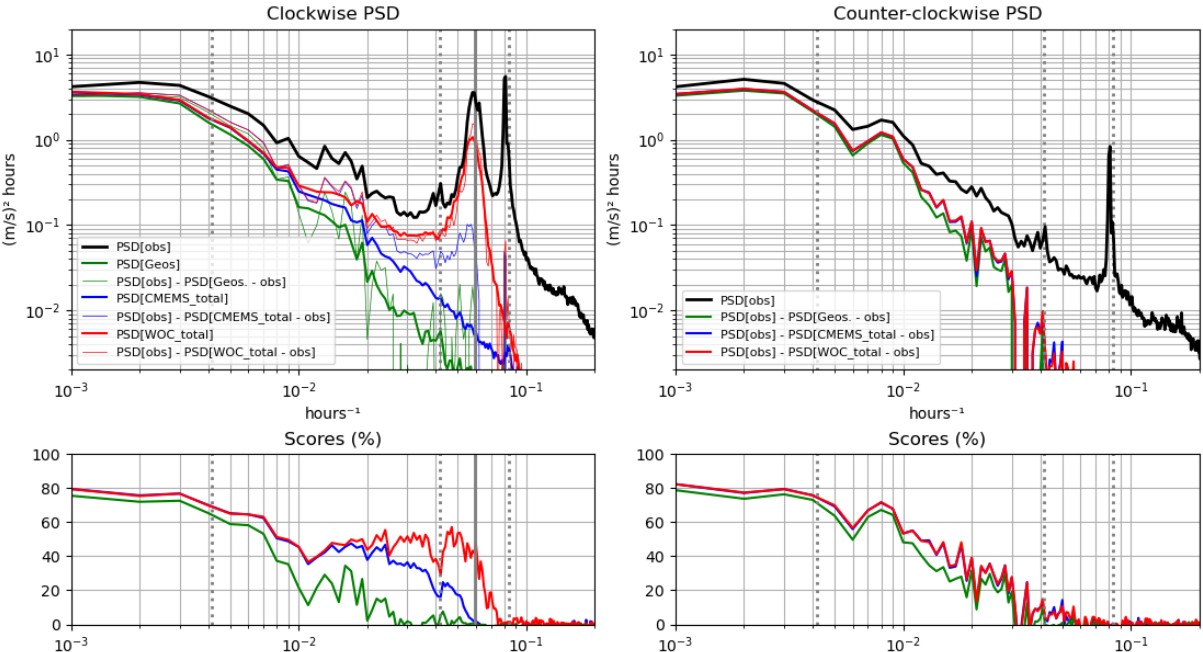

**Figure 4.** Upper-left panel: power spectral density (in the clockwise time-frequency domain between 1000 hours and 2 hours) of the drogued-drifter current observations in the 40°N to 50°N latitude range, in black. The thick-colored curves represent the power spectral densities of the various estimates (geostrophy, CMEMS MOB-TAC and WOC impulse function estimates in green, blue, and red, respectively). The thin-colored curves represent the spectrum of the observations minus the spectra of the difference between the estimation and the observations. Upper-right panel : same, in the counter-clockwise direction. Lower panels : ratio between the thin-colored curves and the black curve of the upper panels, multiplied by 100, representing the percentages of reconstruction (explained variance). The vertical dotted lines indicate the 10 days, 24 hours and 12 hours frequencies, respectively from left to right and the vertical solid line indicates the inertial frequency at 45°N (clockwise only).

Equator, explaining about 1/3 of the variability. This derivation does not provide accurate currents in the meridional direction for which the altimetry contribution is indeed zero near the Equator (right panel). At these low latitudes, the CMEMS MOB-TAC and WOC estimation provide some meaningful signals but still representing less than 20% of the observed variance. At higher latitudes, the zonal and meridional components show similar explained variances for the different estimations. Overall, if we look at the globally-averaged values from Figure 7, geostrophy explains  40% of the surface current variability, and the WOC estimation (blue+pink on the Figure) brings an additional 12% to 14% for the zonal and meridional components respectively. This is significantly above the CMEMS MOB-TAC (blue only on the Figure) that brings 6% to 9% for the zonal and meridional components respectively. This may appear small, but the inertial currents are intermittent and therefore the contribution is certainly much higher at times, particularly following a wind event that triggers inertial oscillations. Nevertheless, there is still a large part of unexplained surface current in the drifters (gray areas on the Figure 7) leaving some room for further scientific investigations that will be discussed in the conclusion section..

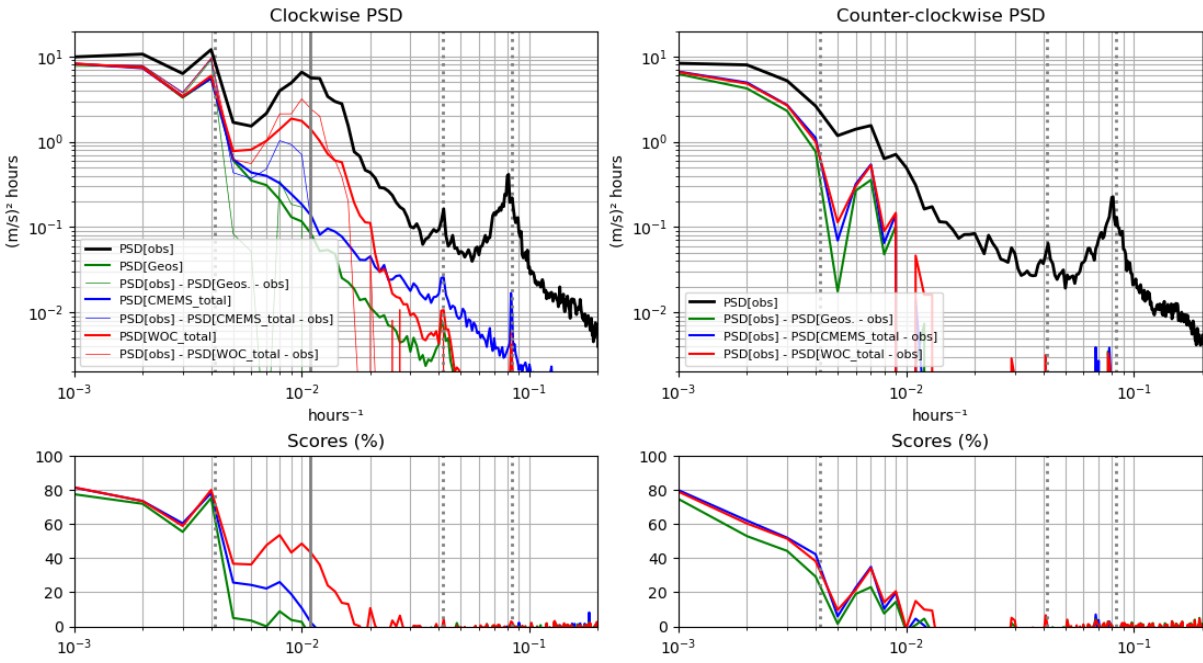

**Figure 5.** Same as Figure 4, averaged between 5°N and 10°N with vertical solid line indicating the inertial frequency at 7.5°N

## 5 Characteristics of the response function

If the WOC method is efficient in capturing some wind-driven current empirically, in particular in the inertial band, it is now
interesting to analyze the features of the response function (i.e. the current response to a wind-stress dirac function), and in
particular its potential variations with the season and the latitude.

Figure 8 represents the response function in blue as a function of time, defined between -1 day and +8 days at different
latitudes and seasons. The real part represents the downwind response and the imaginary part the cross-wind (to the left)
response. For the purpose of this diagnostic, we also computed the response function with the undrogued drifters (in red)
which gives an interesting comparison to the drogued drifters, although they are not used to generate the WOC surface current
product.

First, the values of $G$ are close to zero for negative $t'$, suggesting that the future wind stress is not (significantly) related
to the present current, which is consistent with the fact that ocean currents respond to the wind forcing, rather than the ocean
currents forcing the wind. (We tested a centered window between -8 days and +8 days and also obtained values of $G$ close to
zero for negative $t'$.) However, ocean feedback to the atmosphere obviously exists (e.g., Renault et al., 2016), but this is not
detected in the linear framework of the response function. Then, for positive $t'$, the clear oscillations of $G$ indicate the impact
of the wind history over a few days. These oscillations are close to the inertial frequency (varying with latitude: 14.6 hours at
55°N and nearly 3 days at 10°N) as expected by Eq. 10, with an observed decay.

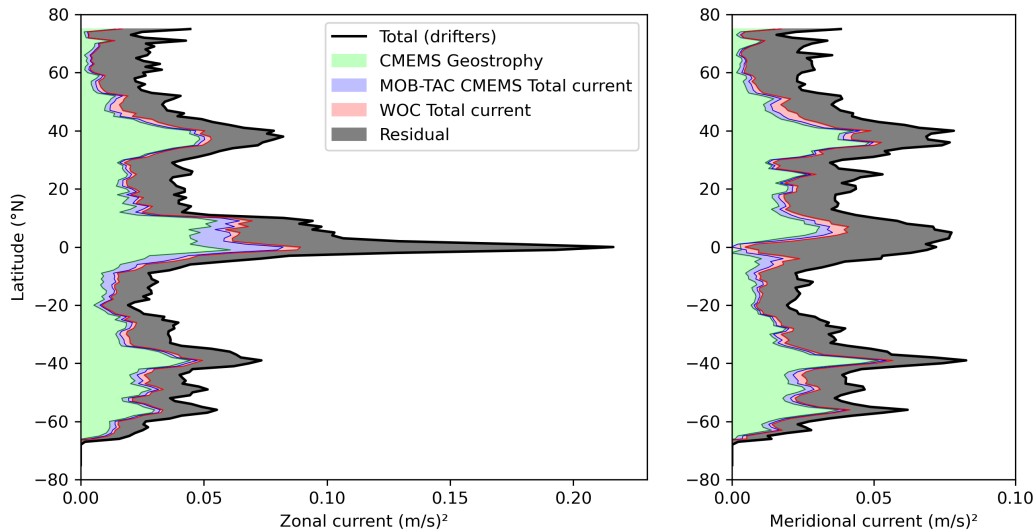

**Figure 6.** Black lines : variance of the observed surface current (zonal component on the left, meridional on the right) as a function of latitude averaged globaly for the drifter database between 2017 and 2020. The green, blue and red lines represent the explained variance of the CMEMS geostrophy, the CMEMS MOB-TAC total current and the WOC total current. The explained variance is defined as the total variance (black) minus the variance reduction after applying the different current estimates. The filled colors indicate the relative amount of additional explained variance between successive products.

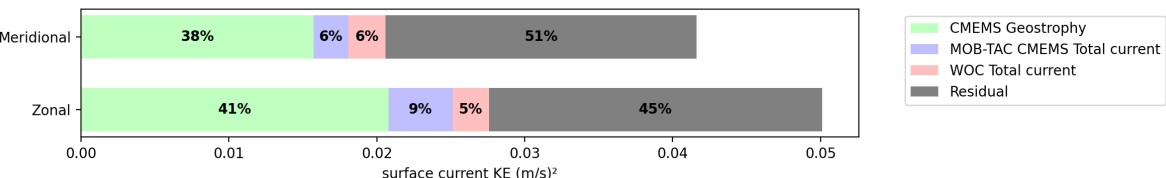

**Figure 7.** Same as Figure 6 averaged globally (area weighted)

The black curve represents a fit for the slab layer response function of Eq. 10. After 12 hours, the slab model, the 15m-drogued-drifter response function, and the undrogued-drifter response functions all show similar behavior. The effect of the seasons at high latitudes is very clear. At 55°N in the winter, the response amplitude after 12 hours is overall twice that of the summer (therefore the thickness of the equivalent slab layer is devided by two). In the tropics, the seasonality is much less pronounced, as expected. We note that the decay rate is quite similar between winter and summer, and is slightly longer at high latitudes than in the tropics. The decay is likely the combination of two effects at least. One is the real attenuation of the NIOs in response to a wind impulse, through energy dissipation or downward energy transfer. A second could be the result of non-linear effects that cannot be captured by the impulse function. For instance, local modifications of the inertial frequency

in response to relative vorticity (e.g. Elipot et al. (2010)) cannot be represented here; we are only resolving the linear response that may attenuate faster than the real inertial oscillations triggered.

Beyond the general similarities with the slab response after 12 hours, some clear and interesting departures from the slab occur in the first few hours of the response function. The departures are observed for the drogued and undrogued drifters in a different manner. The values at short time lags can be interpreted as the result of dynamics occurring right after the wind impulses (typically after the crossing of an atmospheric front). In the following, we speculate on possible interpretations for the observed differences. A first striking feature is the peak of the real part of the function at zero time-lag for the undrogued drifters. This indicates a direct velocity triggered instantaneously in the wind direction. Several effects may explain this peak. First, a surface current that is initially in the downwind direction is, qualitatively, the expected response to impulsive wind forcing; for example, this behavior is clearly seen in the impulse response function derived by Lilly and Elipot (2021) for a specific choice of vertical eddy viscosity (see their Figure 3). A second possible additional effect is wind-slippage that affects primarily undrogued drifters (e.g. Rio et al. (2014), Laurindo et al. (2017)); being undrogued, these drifters are more directly influenced by the wind. The expected response to this "wind slip" would be an immediate response to wind forcing in the direction of the wind, but is not the result of an actual ocean current. A third effect is the Stokes drift from the wind-waves that should also respond rapidly in the wind direction. These three effects likely all play a role in the observed response function, but we do not see an obvious way to disentangle them with the present data. Also, we do not have a clear explanation why the peak seems less pronounced at low-latitudes.

The 15m depth drogued drifters have also interesting departures from the slab in the first few hours. In particular during the winter at high-latitudes. One hypothesis is the presence of temporary re-stratified layers over the very deep mixed layer. This temporary layer would respond to the wind front as a thinner layer in the first hours until the strong mixing (due to the increased wind) transforms the deep mixed layer depth as an active mixing layer, therefore behaving like a slab. In the tropics, we seem to observe an opposite effect at 15m depth with the blue curve reduced in the first  12 hours. This actually might be the result of the same process, but for thinner temporary layers, therefore above the 15m drogue, then destroyed after strong wind impulses.

Although the causes are speculative, this confirms that specific dynamical regimes, fairly different from the slab, are also involved and strongly impact the surface current response to wind stress.

Another representation of the same response functions is represented on Figure 9 along the real and imaginary axes corresponding to the U and V directions respectively. Here, we convolve the response function with a step-function for the wind. This step function, represented by the green arrow along the imaginary axis, is zero for negative time and unitary for positive time. The results, here called the unitary-step response function as represented on the figure, highlight additional features. In particular, the low-frequency response can be directly assessed as being the response to the step function toward infinite time. It corresponds to the point where the curves converge on the figure. This point is to the right of the wind (here in the northern hemisphere) but at a different angle for the drogued and undrogued drifters. The slab-derived step-response functions have constrained angles in the 70°-80° range for the typical values of damping, which is higher than what is fitted for the undrogued and drogued drifters. (As is well known, the form of the damping used in the slab model causes the Ekman transport to be

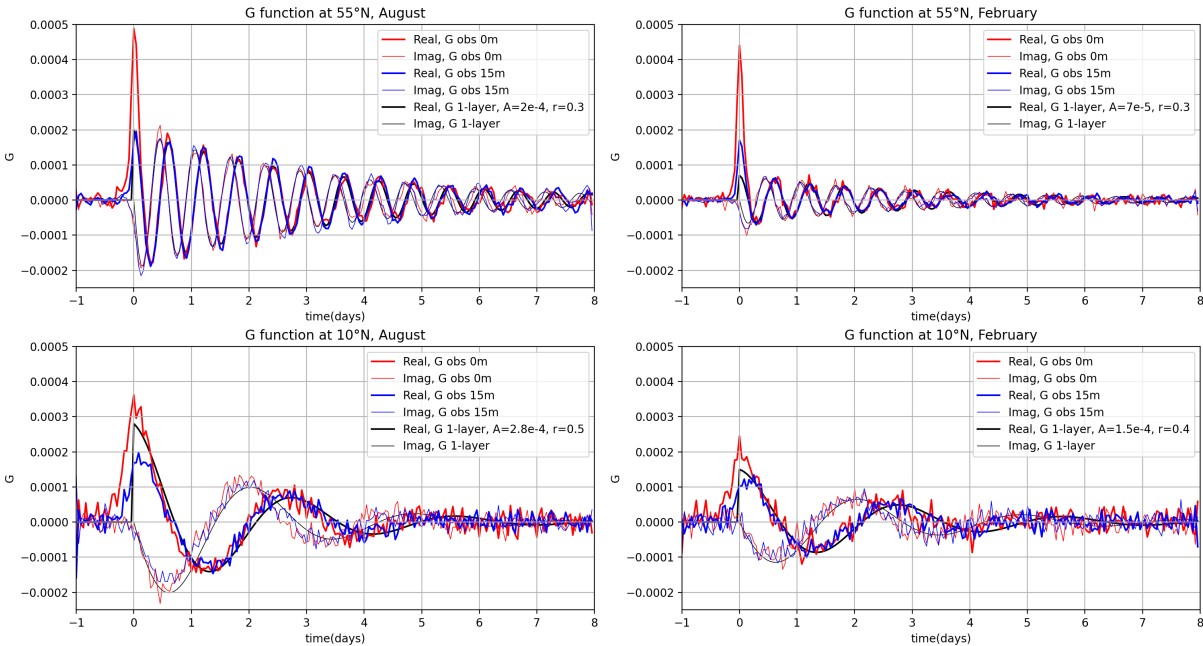

**Figure 8.** Upper panels : G function at latitude 55°N represented as a function of the $t'$ interval, for the real (thick lines) and imaginary parts (thin lines). Upper-left panel in August and upper-right panel in February. Lower panels : same as upper panels, but at 10°N.

slightly less than 90° to the right of the wind.) This again illustrates well the differences and the interest of considering these response function beyond a pure slab dynamic.

## 6 Conclusions and perspectives

This study demonstrated that a purely empirical relation between wind forcing and a large part of the wind-driven surface current can be easily learned from the drifters to provide surface current estimates based on wind stress reanalyses. It provides a potential step forward to the operational total surface current from the CMEMS MOB-TAC based on a similar methodology and input data but here exploiting higher frequencies through the estimation of an impulse response function. The recent accumulation of high-quality drifter data at high-frequency allowed this step forward. The resulting WOC surface current estimates have been successfully validated with independent observations (drifter data not used in the fit of the impulse function) in comparison with the total surface current from CMEMS MOB-TAC. Although the relative gain of explained variance is about 10%, the gain during intermittent near inertial oscillation events is certainly much higher.

The analysis of the response function learned from the drifters may also yield new insights into the physics. Indeed, we found that the similarities with a slab model response are not always true especially in the first few hours of the response. Speculative causes have been discussed, in relation with the existence of temporary layers. The single damping parameter r of the slab layer model is probably unable to capture all the processes leading to energy dissipation and propagation. The longer

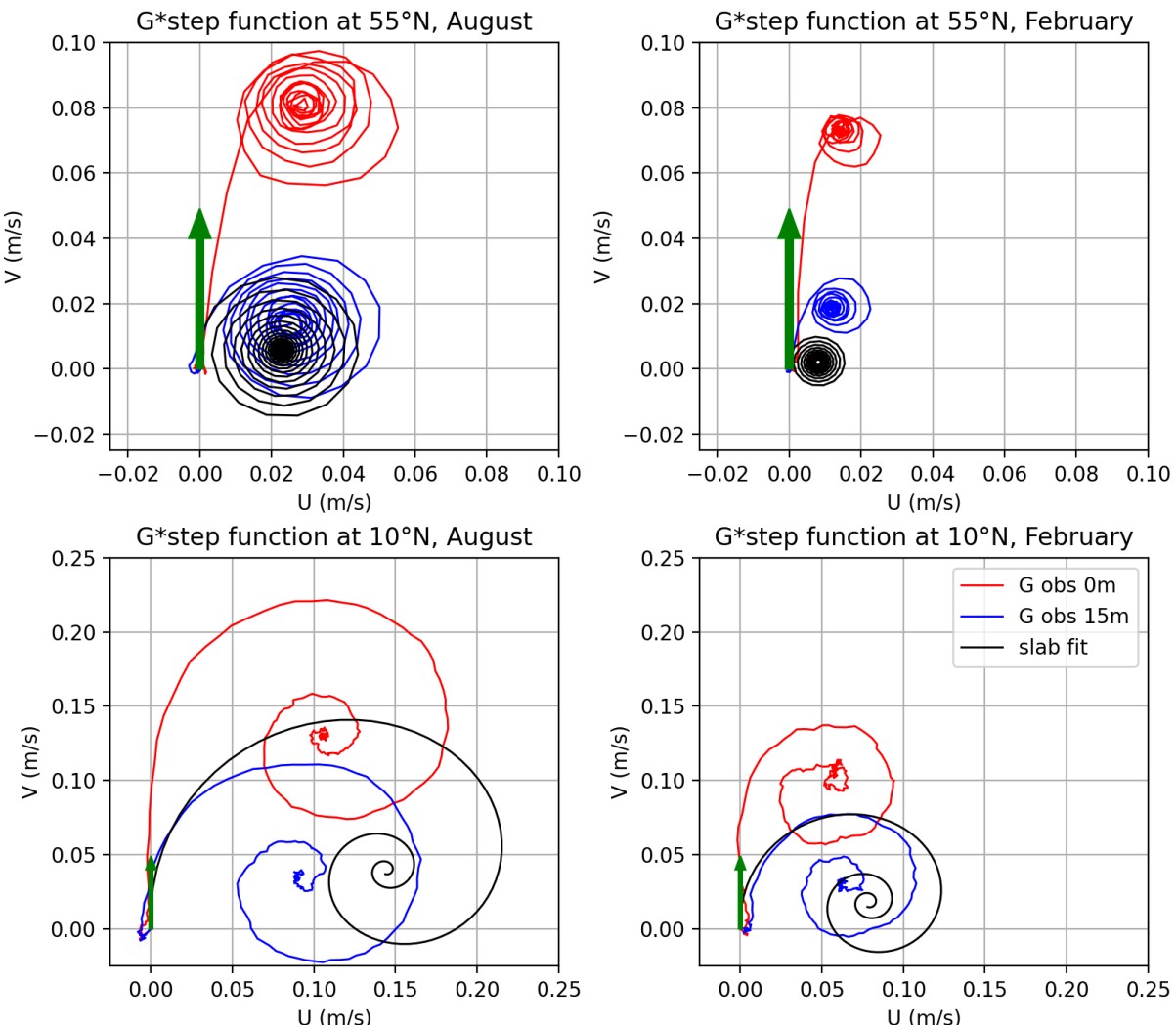

**Figure 9.** Integration of the response function with a unitary-step function of the wind represented by the green arrows. The result, called step-response functions, are represented in the (U,V) plane by the red, blue and black lines for the undrogued drifters, drogued drifters and slab respectively. The four panels represent the different latitudes and seasons as in Figure 8

term (>12h) response is nonetheless similar with a slab response, for both undrogued and 15m-drogued drifters, and with amplitudes clearly related with the seasons out of the tropics (the thickness of the slab being larger in the winter season). This computation of the response function opens the door for considering further dependencies beyond the seasons and latitude to better understand the physical processes in the upper ocean layers in response to the wind. Additional parameters such as the subsurface density profile or sea state may be introduced as parameters in the empirical computation of the response function

(here limited to the meridional and seasonal variations). For instance, diurnal and semi-diurnal processes are known to affect the upper Ocean response (Masich et al. (2021), Cherian et al. (2021), Reeves Eyre et al. (2024)).

These more complex dependencies probably partly explain why a large part of the signal is still unresolved when compared to independent observations. We expect that a lot of progress can be made by considering additional datasets that contain additional information on local sub-surface properties, which could also allow new insights into the physics of the subsurface processes. The wind stress itself may also feature processes not resolved by the wind-stress reanalysis which may also explain another part the remaining signal, as supported by Klenz et al. (2022). We could potentially learn a great deal more about the physical processes in upper ocean from global, coincident measurements of ocean vector winds and ocean surface currents that could be measured from satellites (e.g., Rodríguez et al., 2019) by using a data-driven approach to examine the relationship between the two quantities.

## Data availability

The dataset of surface current generated in this study is available on the World Ocean Circulation ESA project website: https://www.worldoceancirculation.org/

## Author contribution

CU led the study, developed the technical framework and wrote most of Sections 1, 2, 3.2, 4, 5, and 6. TF provided theoretical guidance, contributed to the result interpretation, and wrote Section 3.1. BC proposed the impulse function fitting approach and also contributed to the result interpretation. LG and LGN supported some technical and data management tasks. MHR provided guidance throughout all steps of the study and GD offered advice for the validation of the results presented in Section 4.

## Competing interests

The authors declare that no competing interests are present.

## Acknowledgements

This research was funded by the European Space Agency through the World Ocean Current project (ESA Contract No. 4000130730/20/I-NB ) for sections 1-4, the UpperDyn project (ESA contract No. 1-12306/24/I-EB) for section 5 with complement from the Centre National d'Etudes Spatiales (CNES Contract No. 230376-00 ODYSEA Phase A support). JTF was supported by the US National Aeronautics and Space Administration (Contract 80GSFC24CA067), via a subaward from the University of California San Diego to the Woods Hole Oceanographic Institution.

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
