# Peer review of "A data-driven wind-to-current response function and application to Ocean surface current estimates"

_EGUsphere, 2025_

## Referee Comment (RC1)

Referee comment on "*A data-driven wind-to-current response function and application to Ocean surface current estimates*" 2025, Clément Ubelmann et al.

Submitted by Jack Reeves Eyre
30 April 2025

**Overview**

This manuscript presents an empirical method to calculate a response function of ocean surface current to near-surface wind. Such a function can then be used to estimate the surface current resulting from any arbitrary wind forcing. In this case the response function varies only with latitude and season, but is still able to capture a significant fraction of the current variation observed with drifting buoys.

The paper gives a convincing demonstration that the method serves the intended function. It is mostly well written, with high quality figures. Improvements can be made in the description of datasets used, and in the discussion of some concepts underlying the idea. I recommend publishing after minor revisions.

**Major comments**

**Meaning of wind driven currents**

My main scientific issue with this work is the implicit assumption that wind-driven currents means inertial oscillations forced stochastically by wind events. This is clearly a large part of current total variance, especially in mid- and high-latitudes, and is well captured by the method, as shown by the analysis. However, there are wind-driven current variations at diurnal and shorter time scales that the authors do not discuss, and that this method does not seem to capture very well. This is particularly true of diurnal and semidiurnal variations in the tropics (red line falls below blue line in Figure 5) but is also true at the small diurnal peak in drifter observations in Figure 4.

I do not think more analysis is needed to address this, but these phenomena, and related weaknesses in the method, should be discussed. A couple of concepts that could be mentioned in this light:

- Diurnal warm layers can create significant diurnal wind-forced current variations (e.g., Masich et al, 2021, Cherian et al. 2021, Reeves Eyre et al. 2024) that propagate to 15 m or below. Would adding stratification to the explanatory variables help capture this?
- Diurnal and semidiurnal wind stress variations (e.g., Ueyema and Deser, 2008) could contribute to current variability. However, ERA5 does not use time varying SST, so may

not capture these wind stress variations. Would using a wind stress dataset that does capture these variations give better results in this respect?

- Even at higher latitudes in winter, there may be diurnal variations of wind that could force a current response (Clayson and Edson, 2019). Again, ERA5 may not capture this.

Clayson & Edson: https://doi.org/10.1029/2019GL082826
Ueyema and Deser: https://doi.org/10.1175/JCLI1666.1
Masich et al https://doi.org/10.1029/2020JC016982
Cherian et al https://doi.org/10.1175/JPO-D-20-0229.1
Reeves Eyre et al https://doi.org/10.1029/2023GL104194

**Current datasets**

The references to different current datasets throughout the text are inconsistent and confusing. I note the following datasets/notations, which are mostly used without further explanation or references to other publications or sources:

1. "Globcurrent CMEMS MOB-TAC" (line 10)
2. "(GlobCurrent)[now operational in CMEMS, or the OSCAR dataset]" (Lines 42-43). How many different datasets is this? Is OSCAR another name for the same dataset, or a different dataset?
3. "CMEMS" (line 66). The only one with a reference.
4. "CMEMS Multiobs total zonal current" (Figure 1)
5. "CMEMS Ekman estimate" (Figure 1 caption).
6. "CMEMS Ekman model" (Line 149)
7. "CMEMS MOB-TAC Ekman zonal current" (Figure 4)
8. "Globcurrent/CMEMS (including Ekman)" (Line 163)
9. "Globcurrent/CMEMS" (Lines 167 and several places thereafter)
10. "GEOS" and "CMEMS_total" (Figures 4 and 5)
11. "Altimetry" and "Ekman (MOB-TAC CMEMS)" (Figures 6 and 7)
12. "Geostrophy from CMEMS" and "Globcurrent/CMEMS (geostrophy+Ekman)" (Figure 6 caption)
13. "Operational Globcurrent MOB-TAC CMEMS" (Line 263)
14. "MOB-TAC CMEMS" (Line 267)

I probably could class this as a minor point, but it happens often enough to become a major source of confusion. Please pick a consistent notation and explain it early on. Give references for each distinct source.

**Minor comments**

Lines 57-58: Consider adding section 6 to the section descriptions.

Line 64: Reference to Argo positioning is incorrect. This should be "Argos". Consider also adding a note that this is a distinct technology to ARGO buoys, as some readers may confuse this adjacent technology.

Line 66: Is the "CMEMS" here the same as the "CMEMS-MOB-TAC" mentioned in the abstract and Figure 3? If so, please explain that here. If not, please add a description of what the "CMEMS-MOB-TAC" is.

Figure 1: The time axis on the lower panel is quite unintuitive: if possible, replace with dates. Additionally, further symbols (like the red dot) translating between trajectory (in the upper right panel) and time (in the lower panel) would be helpful. E.g., a different symbol every 10-15 days.

Line 93: Give exact value of T. E.g., "T is set to be xxx hours, or approximately 1 week."

Line 97: Change "studt" to "study".

Lines 98-99: "ageostrophic motions not resolved by altimetry". Aren't the ageostrophic motions, by definition here, those parts of the total current not resolved by altimetry? Should this read "geostrophic motions not resolved by altimetry"?

Line 106: Please define "r".

Sections 3.2 and 3.3: Please take care to ensure consistent bold or regular fonts for characters in the equations and in the text. If the different fonts are meant to be different, please explain the significance.

Line 148: What does "WOC" stand for?

Figure 3: Consider changing the labels of "Unsteady-Ekman zonal current" to WOC, for consistency with the text.

Figures 4 and 5: Consider adding vertical lines corresponding to intuitive frequencies: e.g., at 12-hourly, 24, 48, 120, and inertial for representative latitude. Also, consider y axis labels.

Line 170: Change "barotopic" to "barotropic".

Line 185: Discuss physical significance of counterclockwise compared with clockwise.

Figure 8: Add some kind of depiction of inertial frequency for relevant latitude.

Line 251: Change "Figure 8" to "Figure 9".

Line 254-255: It could be helpful to illustrate this end point concept on one or more of the figures.

Figure 9: It is slightly confusing to refer to the axes as real and imaginary in the caption (and line 251), but label them U and V on the panels' axis labels. Please make them consistent one way or the other.

Line 266: Unclear what "remains overall of 10%" means. Please clarify.

---

## Author Comment (AC1)

*Review by Jack Reeves Eyre:*

*Meaning of wind driven currents*
*My main scientific issue with this work is the implicit assumption that wind-driven currents means inertial oscillations forced stochastically by wind events. This is clearly a large part of current total variance, especially in mid- and high-latitudes, and is well captured by the method, as shown by the analysis. However, there are wind-driven current variations at diurnal and shorter time scales that the authors do not discuss, and that this method does not seem to capture very well. This is particularly true of diurnal and semidiurnal variations in the tropics (red line falls below blue line in Figure 5) but is also true at the small diurnal peak in drifter observations in Figure 4. I do not think more analysis is needed to address this, but these phenomena, and related weaknesses in the method, should be discussed. A couple of concepts that could be mentioned in this light:*
*● Diurnal warm layers can create significant diurnal wind-forced current variations (e.g., Masich et al, 2021, Cherian et al. 2021, Reeves Eyre et al. 2024) that propagate to 15 m or below. Would adding stratification to the explanatory variables help capture this?*
*● Diurnal and semidiurnal wind stress variations (e.g., Ueyema and Deser, 2008) could contribute to current variability. However, ERA5 does not use time varying SST, so may not capture these wind stress variations. Would using a wind stress dataset that does capture these variations give better results in this respect?*
*● Even at higher latitudes in winter, there may be diurnal variations of wind that could force a current response (Clayson and Edson, 2019). Again, ERA5 may not capture this.*

*Clayson & Edson: https://doi.org/10.1029/2019GL082826 Ueyema and Deser: https://doi.org/10.1175/JCLI1666.1 Masich et al https://doi.org/10.1029/2020JC016982 Cherian et al https://doi.org/10.1175/JPO-D-20-0229.1 Reeves Eyre et al https://doi.org/10.1029/2023GL104194 Current datasets*

**We thank you for this important point, which was also raised by the other reviewer. The first version of the draft was not clear enough about what we intended to resolve, and what we did not resolve by design of the method. Following this comment, the whole manuscript has been reworded, to clearly assess that we aim at reconstructing some part of the wind-driven current (the linear response to the wind forcing considered). As you pointed out, the variations of G (constrained to only geographical and seasonal variations) is a limitation that does not allow to capture diurnal/semi-diurnal effects of the response. We had to impose this because of the limited sampling of the drifters. It is now further discussed in the conclusion, with a paragraph mentioning the shorter time scales of the wind-driven current response. In particular for the Tropics where the diurnal signal is probably higher than at mid/high latitudes, we acknowledged the limitations of the method in the discussion of Figure 5.**
**Regarding your mention that the red curve falls below blue when approaching daily scale in the Tropics, it is also interesting to see that at these scales, the scores of all estimations (including total current from CMEMS MOB-TAC in blue) are around zeros. So**

**even the small amount of energy captured has inaccurate phases and is likely unmeaningful at daily wavelength.**

**Significant changes have been made along the manuscript (Introduction, discussion of the Equations section 3, figure description and conclusion) to address these points and in particular to clarify what we resolve and do not resolve.**

*The references to different current datasets throughout the text are inconsistent and confusing. I note the following datasets/notations, which are mostly used without further explanation or references to other publications or sources: 1. "Globcurrent CMEMS MOB-TAC" (line 10) 2. "(GlobCurrent)[now operational in CMEMS, or the OSCAR dataset]" (Lines 42-43). How many different datasets is this? Is OSCAR another name for the same dataset, or a different dataset? 3. "CMEMS" (line 66). The only one with a reference. 4. "CMEMS Multiobs total zonal current" (Figure 1) 5. "CMEMS Ekman estimate" (Figure 1 caption). 6. "CMEMS Ekman model" (Line 149) 7. "CMEMS MOB-TAC Ekman zonal current" (Figure 4) 8. "Globcurrent/CMEMS (including Ekman)" (Line 163) 9. "Globcurrent/CMEMS" (Lines 167 and several places thereafter) 10. "GEOS" and "CMEMS_total" (Figures 4 and 5) 11. "Altimetry" and "Ekman (MOB-TAC CMEMS)" (Figures 6 and 7) 12. "Geostrophy from CMEMS" and "Globcurrent/CMEMS (geostrophy+Ekman)" (Figure 6 caption) 13. "Operational Globcurrent MOB-TAC CMEMS" (Line 263) 14. "MOB-TAC CMEMS" (Line 267) I probably could class this as a minor point, but it happens often enough to become a major source of confusion. Please pick a consistent notation and explain it early on. Give references for each distinct source.*

**This comment was also shared by all reviewers. We apologize for the confusion. In the new version, the datasets are named consistently. Note that the entire dataset section has been re-written. In particular, the three input datasets, our output dataset and comparison datasets are all clearly referred to with accesses and DOI.**
**The figures have also been updated with consistent names**
**This was indeed not a minor point as shown by the number of edits we made. We hope that the present version is clear about the datasets, and that it helps easier understanding.**
**We also find out some typos (ageostrophic versus total current, etc..) that caused confusion in the references to datasets. They should be fixed and clarifications have been added in the dataset description about the distinction between total//geostrophic/ageostrophic currents.**

*Minor comments Lines 57-58: Consider adding section 6 to the section descriptions.*
**Done**

*Line 64: Reference to Argo positioning is incorrect. This should be "Argos". Consider also adding a note that this is a distinct technology to ARGO buoys, as some readers may confuse this adjacent technology.*

**This is right. We modified the paragraph as follows : The surface drifter velocities (estimated jointly from the unevenly distributed observed positions) at hourly frequency are used. Both ARGOS and GPS data are used to allow the 10-year extension of the study, although the GPS data, collected with a different technology, are more accurate (\cite{yu19}) and fairly dominant after 2015.**

*Line 66: Is the "CMEMS" here the same as the "CMEMS-MOB-TAC" mentioned in the abstract and Figure 3? If so, please explain that here. If not, please add a description of what the "CMEMS-MOB-TAC" is.*

**Yes it is the same dataset. As mentioned in the response to the main comments, the dataset references have been clarified. Again we apologize for the confusion.**

*Figure 1: The time axis on the lower panel is quite unintuitive: if possible, replace with dates.*

**Good suggestion, done!**

**We also changed figures 2 and 3 with date labels so now all figures show the time with the same Year/month/day format.**

*Additionally, further symbols (like the red dot) translating between trajectory (in the upper right panel) and time (in the lower panel) would be helpful. E.g., a different symbol every 10-15 days.*

**Thanks , the new figure has three symbols, the first one 8 days before in the quiet-wind period, the second at the front location and the 3rd one 2 days after when the near-inertial signal is strong.**

*Line 93: Give exact value of T. E.g., "T is set to be xxx hours, or approximately 1 week."*
**Done (8 days) and we justified this choice from sensitivity tests (is is actually a trade between actual impulse function variance and noise introduced when too many parameters are fitted)**

*Line 97: Change "studt" to "study".*
**Done**

*Lines 98-99: "ageostrophic motions not resolved by altimetry". Aren't the ageostrophic motions, by definition here, those parts of the total current not resolved by altimetry? Should this read "geostrophic motions not resolved by altimetry"?*

**Yes, indeed! Thanks for the comment. The entire paragraph was modified (also following other comments). Regarding this point, we proposed the following sentence: "(...) additional signal represented by $\epsilon$. $\epsilon$ may contain some missing geostrophy (errors in $u_{g_{obs}}$), errors in the drifter measurement of current, error**

**in the knowledge of $\tau_0(t)$ and any ageostrophic current that would not be captured by the convolution of $G$ with the forcing $\tau_0(t)$."**

*Line 106: Please define "r".*
**Done**

*Sections 3.2 and 3.3: Please take care to ensure consistent bold or regular fonts for characters in the equations and in the text. If the different fonts are meant to be different, please explain the significance.*
**Yes, bold characters were inserted at wrong places in 3.2. Now we use the \vec{} latex notation for all vectors as suggested by the template.**

*Line 148: What does "WOC" stand for?*
**"World Ocean Current" an acronym of an ESA funded project. It is now explained in the dataset section with a reference. Thanks for this obvious missing information!**

*Figure 3: Consider changing the labels of "Unsteady-Ekman zonal current" to WOC, for consistency with the text.*
**Done**

*Figures 4 and 5: Consider adding vertical lines corresponding to intuitive frequencies: e.g., at 12-hourly, 24, 48, 120, and inertial for representative latitude. Also, consider y axis labels.*
**Good suggestion, we added vertical lines for 12h, 24h, 240h and the inertial frequency corresponding to the average latitude.**

*Line 170: Change "barotopic" to "barotropic".*
***Done***

*Line 185: Discuss physical significance of counterclockwise compared with clockwise.*
**Yes, we discussed the physical significance earlier in the section, when the terms clockwise/counter-clockwise are used for the first time.**

*Figure 8: Add some kind of depiction of inertial frequency for relevant latitude.*
**We specified the inertial periods for each of the latitudes on the figure (14 hours and 3 days) so the correspondence with axis units is easier.**

*Line 251: Change "Figure 8" to "Figure 9".*
**Done**

*Line 254-255: It could be helpful to illustrate this end point concept on one or more of the figures.*

**The whole paragraph has been clarified (per request of the 2nd reviewer as well). Now this point is better illustrated, and we explain why this specific representation (step function response) highlights the low frequency response :**
**"In particular, the low-frequency response can be directly assessed as being the response to the step function toward infinite time. It corresponds to the point where the curves converge on the figure. This point is to the right of the wind (here in the northern hemisphere) but at a different angle for the drogued and undrogued drifters."**

*Figure 9: It is slightly confusing to refer to the axes as real and imaginary in the caption (and line 251), but label them U and V on the panels' axis labels. Please make them consistent one way or the other.*
**Indeed, we clarified in the text the U and V directions**

*Line 266: Unclear what "remains overall of 10%" means. Please clarify.*
**We clarified the whole paragraph (same request from the 2nd reviewer): "The resulting WOC surface current estimates have been successfully validated with independent observations (drifter data not used in the fit of the impulse function) in comparison with the total surface current from CMEMS MOB-TAC. Although the relative gain of explained variance is about 10\%, the gain during intermittent near inertial oscillation events is certainly much higher. "**

################################################################################
################################################################################

*Review by Shane Elipot:*

*This paper presents some results of fitting to surface drifter and wind stress data an empirical impulse response function assuming linear dynamics for the forcing of the upper ocean. These results form the basis of a model for unsteady wind-driven near-surface ocean current estimates which are distributed as part of the Globcurrent CMEMS MOB-TAC product.*

*General comments:*

*I think the text of this article is not very clear in terms of what the authors define as wind-driven currents, and this should be clarified. Various terms are used such as inertial oscillations, oscillations, unsteady Ekman currents, (WOC?) etc and it is not clear what the authors mean in each instance (see my detailed comments as an attached PDF). The impulse response function framework effectively extracts from the data the linearly forced response to the wind stress, in a manner which peaks at the inertial frequency, but it is*

*broadband in frequency. I think that "time-varying wind-driven" currents or perhaps "unsteady Ekman" currents may be appropriate? There is maybe a need to further clarify the difference between the wind-driven currents derived in this study and the ones from "Globcurrent/CMEMS".*

**We fully agree with this general comment. All the terminologies have been modified. Now, we clearly state that we empirically aim at capturing some ageostrophy directly (linearly) related to wind-forcing. In the end, this is some part of the wind-driven current, limited to the degrees of freedom for the impulse function G, and also to the physical content of the ERA5 re-analysis wind-stress (certainly missing high-frequency content). It turns out that we capture a large part of inertial-band energy that was not captured by the CMEMS MOB-TAC surface current based on similar input datasets. But not only, and as you correctly state, the response is broadband. We considered different options for the terminology and given these considerations, we decided to employ "some wind driven current" with a clear mention that it is limited to the linear response part (and even the part of it that can be captured with the method ).**

**As for the various terms in the datasets, we apologize for the lack of clarity and the confusions brought by different terms designing the same datasets. In the new version, the dataset are clearly mentionned, referred, and pointed in the text with unique terminology.**

*I think the paper is a little bit light on the description of the calculations conducted in sections 3.2 and 3.3. As an example, I am not quite sure how the data were divided in time and in latitude (daily bins? 1 degree latitude bands?). In order for others to try to replicate your study, I believe you could provide more information. In particular, have you conducted any sensitivity on the length of the wind stress time series input? Is T= 10 days an optimal value? Could it depend on latitude and season?*

*In general, the mathematical notations could be made more rigorous. As an example, some symbols are written sometimes with bold fonts and sometimes without (examples in sections 3.2 and elsewhere). More importantly, I suggest to revise the formulation of the impulse response function in order to make clearly appear the convolution operation and to correctly deal with the bivariate nature of the various quantities. Examples can be found in*

*Chapter 6 of Bendat and Piersol (cited by the authors) as well as in Elipot and Gille (2009) and Lilly and Elipot (2021).*

**Yes, we agree that technical elements were missing; The section "Resolution of the inverse problem to fit G" has been re-written with details such as the seasonal harmonic to fit G (we do not bin the data in time but fit an harmonic), and the latitude bands (that are indeed 1° wide).**

**Yes, the choice of a 8-day time window was the result of a sensitivity analysis, now mentioned in the manuscript. This is a trade between considering potentially more physics, but also introducing over-fitting given the limited amount of drifter data. The sensitivity test was performed on independent drifter data to assess the overfitting effect. The question of the optimal time-window that could vary with latitude and seasons is a very good one, now discussed in the manuscript. It is certainly related to the damping rates (as for the equivalent SLAB dynamics) and more generally to the time before momentum fluxes dissipate the upper layer energy. But also to the magnitudes of the target signal versus the magnitude of the unresolved signal (epsillon in the new set of equations). For example, if a region has dominant and strong wind-driven currents that are linearly related to ERA5 wind-stress, with low eddy variability to mitigate potential non-linear effects, it is likely that a longer window would help. We acknowledged that considering such variations would be one of the possible improvements of the method.**

**The section 'rationale for an impulse function' has been entirely re-written ; The justification for an impulse function is, we believe, more rigorous in the new version. The steps (and different assumptions) leading to the linearization of the current with respect to the wind forcing are now detailed (we also discuss the possible non-linearities that would not be captured). Also, the integration operation is redefined to be consistent with the definition of the convolution (note that we changed the sign of t').**

**Indeed, the bivariate nature and its description with complex-valued quantities were not clear. Now we introduce the complex values earlier (after Eq.4) with a more rigorous definition (inspired from Lilly and Elipot 2021).**

**Finally, the mathematical characters have been reviewed (all vectors with the latex \vec{} notation as suggested by the template).**

*Some more detailed (but not necessarily minor) comments are attached in a separate PDF document.*

**We thank you very much for all the notes in your attached PDF document. The comments were extremely useful to the new version of the manuscript. All of them were taken into account with appropriate modifications of the manuscript and figures. Below, we provide responses to some of these comments that contain specific questions or important points to answer..**

*Number: 5 Author: Subject: Comment on Text Date: 5/19/25, 1:37:25 PM If you are using the hourly product distributed by AOML as referenced in Elipot et al. 2016, then note that the hourly and velocity estimates are estimated jointly from the unevenly distributed observed positions. Velocity are not simply derived from the original positions.*
**Yes we used the hourly product and we mentioned that velocities are not simply derived from the original positions.**

*Number: 7 Author: Subject: Comment on Text Date: 5/19/25, 1:46:53 PM This may not be very important because the geostrophic velocity is mostly low-frequency, but do you know if the time stamp of the geostrophic velocity maps is at noon or midnight? That's a 12-h difference in the interpolation scheme.*
→ **Yes, the time stamp of the CMEMS geostrophic current is 00H UTC. We applied the time-linear interpolation scheme accordingly**

*Number: 9 Author: Subject: Comment on Text Date: 5/19/25, 1:53:59 PM inertial currents, oscillations, unsteady-Ekman response: are you conflating all these terms? I think you should be a bit more clear. Could there be any tidal component in this particular example?*
**The oscillations are very clear both on the drifter trajectory, and the derived zonal current shown on the bottom panel. Although they may combine several effects possibly including tidal signals, we think they are mostly inertial signal that could be reconstructed from the wind forcing. This gives some confidence on the reliability of the datasets to explore the wind-driven current response**

*Number: 6 Author: Subject: Comment on Text Date: 5/19/25, 2:29:46 PM rms of what?*
**Misfit with the observation**

*xt Date: 5/19/25, 2:35:08 PM consider using cyclonic and anticyclonic?*
**Yes, we mentioned the anticyclonic motion corresponding to the clockwise direction in the northern hemisphere**

*Number: 2 Author: Subject: Comment on Text Date: 5/19/25, 3:11:19 PM Have you conducted some sensitivity experiments on the value of T? Is T=10 days the value that maximizes the explained variance globally?*
**This is a very good point. Now detailed in the previous section about the implementation : Some sensivity tests have been conducted to find an optimal time extension, based on the maximum of explained variance over independent drifter data. Globally, the optimal was around 8 days, which is certainly a compromise between the actual extension of $G$**

**(the wind-driven linear response time) and possible overfitting due to the limited amount of drifter data.**

*Number: 5Author: Subject: Comment on Text Date: 5/19/25, 3:14:11 PM Section 2 seemed to imply that you used only drogued drifter data?*

**Yes, we used undrogued drifter data only here for the purpose of this diagnostic, but not for the rest of the study. We clarified in the text : "For the purpose of this diagnostic, we also computed the response function of the undrogued drifters (in red) which gives an interesting comparison, although they are not used for the rest of the study to generate the WOC surface current product. "**

*Number: 1Author: Subject: Comment on Text Date: 5/19/25, 3:01:22 PM Maybe you could display the x axis on a logarithm scale?*

**This is an interesting suggestion, but after applying it (see below) we preferred to keep the linear scale. Even if the low-variability region can be harder to read with linear scale, we think it is easier to read the number.**

[Figure]

*Number: 2Author: Subject: Comment on Text Date: 5/19/25, 3:26:37 PM Are you sure? I think that if you have a unit wind stress forcing function, doubling the H parameter of the slab layer model will reduce, not increase, the surface current.*

**Correct!**

*Number: 3Author: Subject: Comment on Text Date: 5/19/25, 3:27:12 PM What's inertia here? I think you can simply say the "response".*

**Thanks for the good suggestion**

*Number: 4Author: Subject: Comment on Text Date: 5/19/25, 3:29:44 PM This was studied extensively from drifter data in Elipot et al. 2010 "Modification of inertial oscillations by the mesoscale eddy field".*
**Indeed, we added the reference, thanks**

*Number: 1Author: Subject: Comment on Text Date: 5/19/25, 3:32:24 PM both drogued and undrogued drifters are expected to be affected, but undrogued more strongly. See Laurindo et al 10.1016/j.dsr.2017.04.009*
**Indeed. We also added Laurindo et al. paper as a reference.**

*Number: 2Author: Subject: Comment on Text Date: 5/19/25, 3:49:46 PM Not sure why it is an equivalent step response. It is a plot of the hodograph of the wind-driven response to a unitary step wind stress forcing.*
**We clarified the few sentences regarding this point, as follows:**
**The results, here called the unitary-step response function as represented on the figure, highlight additional features. In particular, the low-frequency response can be directly assessed as being the response to the step function toward infinite time. It corresponds to the point where the curves converge on the figure.**

---

## Referee Report (RR1)

Referee comment on "A data-driven wind-to-current response function and application to Ocean surface current estimates" 2025, Clément Ubelmann et al.

Submitted by Jack Reeves Eyre 29 August 2025

**Overview**

This manuscript presents an empirical method to calculate a response function of ocean surface current to near-surface wind. Such a function can then be used to estimate the surface current resulting from any arbitrary wind forcing. In this case the response function varies only with latitude and season, but is still able to capture a significant fraction of the current variation observed with drifting buoys.

The paper is a significantly amended version of an earlier submission, and the changes have improved the paper. In particular, I commend the authors on much clearer dataset descriptions, as well as clearer mathematical notation and improved physical interpretation. I recommend publishing after minor revisions.

**Minor comments**

Line numbers refer to the version with tracked changes.

Line 36: Please say what kind of application "(e.g. Shrira and Almelah, 2020)" is.

Line 37: "... but also for practical and societal...". I suggest this should be a new sentence, to avoid a long and slightly awkward sentence.

Line 99: Unclear what "hourly mean and output frequency" means. Please clarify.

Figure 2: Y-axis units needed.

Lines 233-234: I would recommend adding another equation to spell out "from the adjoint of the linear operations in Eq. 13 and from the adjoint of the convolution Eq. 6."

Line 245: A word is needed between "phase" and "quite accurate", or rearrange the word order.

Line 272: "Except for the CMEMS MOB-TAC...inertial band:" I think you are saying here that the CMEMS MOB-TAC is an exception from the previous sentence ("phases ...are correct"). However, it is quite ambiguous because it isn't a full sentence. I would suggest something like "One exception to this is the CMEMS MOB-TAC... inertial band:"

Line 289" "short time wavelength" is a confusing phrase. Do you mean "short period"?

Line 293: "proposed" does not seem like the right word here.

Figure 7 caption: Is the averaging simple (linear in latitude) or area-weighted?

Figure 8 caption: Seems to be missing description of upper right panel. Some detail on the different lines would be helpful here (I know it is mentioned in the text elsewhere). The description of the lower panels does not seem consistent with the figure. The real-imaginary plane does not seem to be part of this figure. Maybe a copy-paste error?

Line 366: "... some wind-driven surface current...". "Some" is a bit vague here – does it mean in some places/seasons, or some component of?

Line 386: Might be nice to remind readers of the "reduced space" here - maybe like "reduced space (capturing only latitude and seasonal cycle)...".